# Integration of Gold Nanoparticles into Crosslinker-Free Polymer Particles and Their Colloidal Catalytic Property

**DOI:** 10.3390/nano13030416

**Published:** 2023-01-19

**Authors:** Jian Hou, Bin Li, Wongi Jang, Jaehan Yun, Faith M. Eyimegwu, Jun-Hyun Kim

**Affiliations:** 1School of Intelligent Manufacturing, Luoyang Institute of Science and Technology, Luoyang 471023, China; 2Henan International Joint Laboratory of Cutting Tools and Precision Machining, Luoyang Institute of Science and Technology, Luoyang 471023, China; 3Department of Chemistry, Illinois State University, Normal, IL 61790-4160, USA

**Keywords:** gold nanoparticle, poly(N-isopropylacrylamide), cross-linker free particle, homocoupling, long-term stability

## Abstract

This work demonstrates the incorporation of gold nanoparticles (AuNPs) into crosslinker-free poly(N-isopropylacrylamide), PNIPAM, particles *in situ* and the examination of their structural and catalytic properties. The formation process of the AuNPs across the crosslinker-free PNIPAM particles are compared to that of crosslinked PNIPAM particles. Given the relatively larger free volume across the crosslinker-free polymer network, the AuNPs formed by the *in situ* reduction of gold ions are detectably larger and more polydisperse, but their overall integration efficiency is slightly inferior. The structural features and stability of these composite particles are also examined in basic and alcoholic solvent environments, where the crosslinker-free PNIPAM particles still offer comparable physicochemical properties to the crosslinked PNIPAM particles. Interestingly, the crosslinker-free composite particles as a colloidal catalyst display a higher reactivity toward the homocoupling of phenylboronic acid and reveal the importance of the polymer network density. As such, the capability to prepare composite particles in a controlled polymer network and reactive metal nanoparticles, as well as understanding the structure-dependent physicochemical properties, can allow for the development of highly practical catalytic systems.

## 1. Introduction

Nanoscale gold particles have shown attractive catalytic properties in various chemical reactions, including oxidation, reduction, dye degradation, and coupling, at mild and ambient conditions [1,2,3,4,5]. Gold nanoparticles (AuNPs) as a catalyst generally exhibit high selectivity but relatively low reactivity compared to conventional noble metals [6]. As such, much effort has been devoted to promoting their reactivity by controlling the structural features of AuNPs, which have shown great promise as a colloid-based catalyst. For example, regulating the size and shape of AuNPs could improve the overall catalytic performances during chemical reactions [2,6]. Specifically, structurally-tuned AuNPs with smaller sizes and multiple defects could induce effective interactions with reactants [7,8,9,10]. Similarly, designing compositionally diverse alloy NPs (e.g., bimetallic and trimetallic) by introducing other noble metals (e.g., palladium or platinum) could also improve both catalytic reactivity and selectivity [11,12,13,14]. However, even after the successful preparation of these gold-based NPs, maintaining the long-term colloidal stability and dispersity for solution-based chemical reactions is still a challenging task. To render enhanced stability to colloidal AuNPs, various host substrates have been employed to efficiently uptake AuNPs by controlling the degree of interactive forces [5,8,15,16,17,18,19].

Integration of catalytically active AuNPs into a solid support structure could secure the long-term stability, which can also allow for the easy recovery and recycling of the composite materials as a colloidal catalyst. As such, various solid substrates have been actively tested as a host material to integrate guest AuNPs and examine their overall catalytic reactivity and selectivity [1,5,15,20,21,22]. Polymer particles are of particular interest due to several advantages, including a tunable surface polarity, functional groups, dispersion capability in various solvents, structures, and stimuli-responsiveness (i.e., pH and temperature) [14,23,24,25,26,27]. The successful preparation of these composite particles allows for investigating the regulation of the reactants’ accessibility to the catalytically active sites of the AuNPs for the design of multifunctional chemical reaction systems. With these unique features, the utilization of relatively porous polymer networks could result in the integration of structurally tunable AuNPs with enhanced catalytic properties for the development of an interesting nanoscale reactor.

One commonly employed porous polymer material in catalytic systems is poly(N-isopropylacrylamide), PNIPAM, particles possessing discontinuous swelling and deswelling properties near the lower critical solution temperature (LCST) [1,3,20,28]. These polymer particles have been extensively fabricated with various crosslinkers and co-monomers to control the formation of nanoscale metal particles and their catalytic functions. In addition, the PNIPAM-derived materials with new functional groups and variable repeating units can easily alter their physicochemical properties. The *in situ* integration of guest AuNPs into these host PNIPAM particles could result in the preparation of metal core-polymer shell type composite materials with great stability [3,22,29]. In chemical reactions, the accessibility of the incorporated AuNPs possessing active sites through the polymer network is an important factor for the catalytic conversion of organic moieties to target products (e.g., mass transfer environment for reactants and products) [20,30,31]. Our previous work also demonstrated the structural behavior of composite materials derived from physically integrated AuNPs into swollen crosslinked PNIPAM particles, as well as their subsequent catalytic applications [30,32,33,34]. Specifically, the reduction of a metal precursor (i.e., gold ions) from charged ions to a metallic state to induce the nucleation of metal clusters. Subsequent coalescent growth via Ostwald ripening results in the formation of colloidal NPs, where this growth is highly influenced by the Brownian motion in the solution. Although the polymer matrix does not possess special functional groups to induce electrostatic and/or covalent interactions with AuNPs, these interactions still limited the controlled formation of guest AuNPs during the *in situ* reduction method, where the degree of interactions and/or the density of the polymer network highly affected the nucleation and growth of the guest NPs [2,30]. Although the resulting composite particles displayed good stability and dispersity, the degree of crosslinking density somewhat deteriorated the mass transfer capability of molecular species (i.e., diffusion rate) in chemical reactions. As such, the use of a relatively loose polymer network could establish a hitch-free mass transfer environment capable of promoting better interactions between the catalytic sites and reactants. 

Here, we took a further step to utilize crosslinker-free host PNIPAM particles whose polymer network could minimally interact with guest AuNPs during the *in situ* formation. The polymer particle network without crosslinkers could then offer an effective diffusion process for gold ions and reducing agents that can result in the controlled formation of AuNPs (e.g., size and distribution). The integrated AuNPs across a highly porous polymer network and free volume could exhibit increased catalytic performance due to their improved stability and dispersion, as well as high mass transfer capability for molecular species during the reaction. In this study, we experimentally examined the formation and integration efficiency of guest AuNPs across crosslinker-free PNIPAM particles. Interestingly, controlling the size of the integrated AuNPs was limited within the PNIPAM particles prepared without using a crosslinker (i.e., highly loose polymer network). This observation could evidently explain the presence of inevitable interactions, including van der Waals, hydrogen bonding, and dipole-dipole interactions between the polymer chains and AuNPs [35,36]. However, the resulting composite particles still maintained great stability and dispersity under various conditions. For practical applications, the composite particles derived from the crosslinker-free PNIPAM were tested as a catalyst in the homocoupling reaction, and this outcome was thoroughly compared to the crosslinked system to understand the overall role of polymer network density in the catalytic reaction. Investigating the integration efficiency and structural feature of reactive metal NPs and the role of the porous polymer particle network in chemical reactions could lead to the development of ideal catalytic systems.

## 2. Materials and Methods

### 2.1. Materials

Ammonium persulfate (APS, >98%), potassium persulfate (KPS, >99%), *N,N’*-methylenebis-acrylamide (BIS, 99%), nitric acid, hydrochloric acid (HCl), trisodium citrate dehydrate (99%), potassium carbonate (K_2_CO_3_, >99%), potassium hydroxide (KOH, 99.98%), phenylboronic acid (>98%), biphenyl (99%), absolute ethanol (EtOH), and hydrogen tetrachloroaurate trihydrate (HAuCl_4_∙3H_2_O, 99.995%) were obtained from Fisher Scientific. N-isopropylacrylamide (NIPAM, 99%, Aldrich) was recrystallized in hexanes and dried under vacuum prior to use. All glassware was cleaned with a strong acid solution and a base bath. The water used in all reactions was obtained from a Nanopure water system (Barnstead/Thermolyne). 

### 2.2. Preparation of Crosslinker-Free and Crosslinked PNIPAM Particles and AuNP-Integrated Composite Particles

Conventional radical polymerization was employed to prepare crosslinker-free and crosslinked PNIPAM particles with and without using the BIS crosslinker. For the preparation of crosslinked PNIPAM particles, the NIPAM monomer (1.00 g, 8.84 mmol), APS initiator (0.12 g, 0.53 mmol), and BIS crosslinker (0.07 g, 0.45 mmol) were completely dissolved in a 500 mL round bottom flask containing 200 mL water. The mixture was purged with ultra-high purity argon gas for 1 h and then heated to 70 °C in an oil bath for 5 h. After cooling to room temperature, the milky white solution containing PNIPAM particles with the 5 mol% BIS crosslinker was filtered through a Whatman grade 1 filter paper. Similarly, the crosslinker-free PNIPAM particles were prepared with KPS initiator (0.14 g, 0.52 mmol) in the absence of the BIS crosslinker under the same reaction conditions. 

To prepare the composite particles, an aliquot of crosslinked or crosslinker-free PNIPAM particle solution (10 mL) was mixed with 1.0 mL of 1 wt% HAuCl_4_^.^3H_2_O in a tall glass vial for 30 min. After introducing 1.0 mL of 1 wt% trisodium citrate solution, the vial was exposed to a solar simulated light (~100 mW/cm^2^) for 3 h to convert gold ions to AuNPs *in situ.* Although the reducing agent could convert the gold ions to AuNPs in the absence of a light source, the reduction process is typically much slower and incomplete to form highly polydisperse AuNPs without light irradiation [37,38]. As such, this light-induced reduction method effectively resulted in the formation of AuNPs in the presence of the polymer particles [21,28,30]. The fast and efficient formation has been explained by the stimulation of surface plasmon resonance (SPR) of initially formed AuNP seeds to accelerate their growth under light irradiation. In addition, the reaction progress was easily monitored by the color transition from light yellow to red although the vial was kept in a water-jacketed beaker (<22 ℃) the entire time. The purified composite particles were then resuspended in EtOH after the removal of free AuNPs, small particles, and unreacted species by centrifugation (4000 rpm for 20 min × 2). 

### 2.3. Homocoupling of Phenylboronic Acid in the Presence of Composite Particles

The purified composite particles in EtOH (2.0 mL) were placed in a glass vial containing the phenylboronic acid reactant (21 mg, 0.17 mmol) and inorganic KOH base (5.6 mg, 0.1 mmol). The reaction was then carried out in a pre-heated oil bath at 65 ℃. An aliquot of the reaction mixture (0.5 mL) was taken out at designated time intervals and centrifuged at 4000 rpm for 20 min to collect the top layer for gas chromatography (GC) analysis. It is noted that the crosslinker-free composite particles used in the reaction were two different amounts of integrated AuNPs (i.e., the as-prepared and concentrated composite particle solutions containing AuNPs that were equal to the crosslinked composite particles).

### 2.4. Characterization

The structural features, including the size, shape, and distribution of the PNIPAM and composite particles, were examined using scanning electron microscopy (SEM, FEI-Quanta 450 operating at a voltage of 20 kV) and transmission electron microscopy (TEM, Hitachi H8100 operating at a voltage of 200 kV, Tokyo, Japan). The samples for the SEM analysis were placed on silicon wafers and coated with a thin layer of gold using a sputter coater (DESKII, Denton Vacuum Inc., Moorestown, NJ, USA). The samples for the TEM analysis were dried on carbon-coated copper grids. ImageJ software (v1.45s, National Institute of Health, Bethesda, MD, USA) was used to examine the size and distribution of the polymer particles and integrated AuNPs by counting over 100 particles. The hydrodynamic diameter and polydispersity of the particles were examined with dynamic light scattering (DLS, ZetaPALS, Brookhaven Instruments Corp., Holtsville, NY, USA) as a function of temperature. The particle samples were diluted either in water or EtOH prior to the measurement of 1 min. All presented data are the averages of at least three measurements. The integrated AuNPs across the crosslinker-free and crosslinked PNIPAM particles were monitored by a UV-visible spectrometer (Agilent, Santa Clara, CA, USA). The purified composite particles were diluted and transferred to a quartz cell, and were scanned at the 200–1100 nm wavelength. The amount of integrated AuNPs across the PNIPAM particles was examined by atomic absorption spectroscopy (AAS, AAnalyst 200, Perkin Elmer, Waltham, MA, USA) equipped with an Ag-Au hollow cathode lamp. An aliquot of the purified composite particles (0.1 mL) was suspended in 1.0 mL of strong acid (1:1 volume ratio of HCl to HNO_3_) for 10 min to completely dissolve the integrated AuNPs. After the addition of 8.9 mL of pure water, the resulting mixture was centrifuged at 4000 rpm for 30 min to remove the PNIPAM particles. The purified solution was subjected to the AAS measurements, where the absorbance was compared to those of gold standard solutions via the Beer−Lambert law. The yield of homocoupling reaction was obtained using gas chromatography (GC, Thermo Focus GC installed with a fused silica capillary column and an FID detector). A temperature programming method (oven temperature from 130 °C to 250 °C with a ramping rate of 25 °C/min under 10 psi pressure) was used to determine the yield of biphenyl products. 

## 3. Results

### 3.1. Formation of Crosslinker-Free and Crosslinked PNIPAM Particles 

Based on four main approaches, the *in situ* synthesis method involves the reduction of metal ions in the presence of preformed polymeric materials, which readily allows for the formation of metal NP-polymer composite particles [30,33,39,40,41]. Our method utilizing a mild reducing agent and visible light irradiation demonstrated the effective incorporation of AuNPs into PNIPAM particles at room temperature, where the fully swollen polymer network of PNIPAM particles was maintained during the formation and growth of AuNPs. Given the absence of strong chemical and electrostatic interactions, the reduction of gold ions to metallic gold clusters/growth via Ostwald ripening resulted in nanoscale gold particles that were physically incorporated across the crosslinked PNIPAM particles below the LCST. The colloidal composite particles then exhibited great stability in various organic solvents and surprisingly high catalytic properties in the homocoupling of arylboronic acid derivatives. Unlike tuning the size of bare AuNPs by controlling the ratio of gold ions and reducing agent, regulating the size and distribution of the incorporated AuNPs across the crosslinked PNIPAM particles was experimentally limited [2,30]. This noticeable limitation could possibly be due to the crosslinking density of the polymer network, restricting the structural control of the forming AuNPs. In this sense, PNIPAM particles without crosslinkers were prepared to serve as host materials during the same reduction of gold ions to form AuNPs. The formation process of AuNPs and stability of the resulting composite particles were also evaluated to understand the influence of the PNIPAM network. Furthermore, the composite particles were tested as a reactive catalyst in the homocoupling of phenylboronic acid to demonstrate their potential use in chemical transformation reactions. 

Initially, crosslinker-free PNIPAM particles were prepared following the modified method using the KPS initiator [42,43,44,45]. This synthetic approach has shown the possibility of forming a globular shape in the absence of crosslinking agents, where KPS can abstract hydrogen atoms from the NIPAM backbone to form a self-crosslinked network. The resulting PNIPAM particles possess a loose and/or hyperbranched polymer network to serve as a host material to incorporate guest AuNPs for colloidal catalytic applications. The overall physicochemical properties of the crosslinker-free PNIPAM particles were examined and compared to the crosslinked PNIPAM particles prepared with 5 mol% BIS. The crosslinker-free PNIPAM particles appeared to be almost clear at room temperature, while the crosslinked particles were slightly milky white. In addition, a much smaller amount of PNIPAM particles was precipitated compared to the crosslinked particles, which was likely due to the high hydrophilicity and loose polymer network, confirmed by their typical extinction patterns (Appendix A). The crosslinked PNIPAM particles with a dense polymer network were scattered notably across the UV and visible wavelength areas. In contrast to the crosslinked PNIPAM particles, SEM and DLS measurements showed a detectably smaller and more polydisperse distribution of the crosslinker-free polymer particles (Figure 1). A much smaller size of the crosslinker-free PNIPAM particles was observed by SEM, possibly due to the significant volume reduction of a loose polymer network (i.e., complete dehydration under vacuum). It is also noted that the crosslinker-free PNIPAM particles with the loosely self-crosslinked and/or hyperbranched polymer network generally displayed irregular shapes and a large distribution. Overall, our radical polymerization with KPS readily resulted in the formation of globular structures.

To examine the swelling and deswelling behaviors of the polymer particles, DLS measurements were carried out as a function of temperature (Appendix A). The crosslinker-free particles displayed the higher swelling and deswelling ratio of ~4.2 with a slightly sharper phase transition nearer to the LCST than the crosslinked particles, which strongly suggested a loose polymer network. Given the larger size variations of the polymer particles, the polydispersity values were also high across the entire temperature range. The crosslinked polymer particles with the lower ratio of diameter changes (i.e., ~2.8) were expected to possess a relatively denser polymer network and smaller free volume across the PNIPAM particles. 

### 3.2. Incorporation of AuNPs into Crosslinker-Free and Crosslinked PNIPAM Particles

After understanding the structural and physical properties of the host PNIPAM particles, the formation of AuNPs was monitored as a function of time. As the reaction was carried out in a water-jacketed beaker, the polymer particles maintained a fully swollen state during the *in situ* formation and growth of AuNPs. It was evident that the reduction of gold ions was much faster in the presence of the crosslinker-free polymer particles where the reaction could be completed in 2 h (Appendix A). However, the total amount of gold ions consumed during the entire reduction process was comparable, which was determined by the color of the final solution and surface plasmon resonance (SPR) pattern (i.e., the intensity ratio of 470 nm to 535 nm absorbance) (Appendix A). Upon purification of the prepared composite particles, the crosslinker-free PNIPAM particles seemed to lose more AuNPs than the crosslinked polymer particles, probably due to the loose polymer network (Appendix A). This observation was evident by the degree of separation after the first centrifugation step (e.g., a dark purple top solution for the crosslinker-free composite particles vs. a light pink top solution for crosslinked composite particles) and final solution color (e.g., purple vs. dark purple), as well as the SPR intensity of the composite particles.

After the removal of free AuNPs, SEM and TEM images were collected to observe the distribution of the composite particles and the incorporation of AuNPs across the PNIPAM particles (Figure 2 and Appendix A). Under the same ratio of gold ions to trisodium citrate reducing agent, the integrated AuNPs on the crosslinker-free particles were slightly larger and polydisperse. In addition, detectably fewer number of AuNPs were integrated around the polymer particles, which strongly supported our observation during the purification process. The DLS analysis also showed a larger distribution of the overall composite particles prepared with the crosslinker-free PNIPAM particles caused by the uneven surfaces and wider variations of the host polymer particles. The integrated AuNPs across the crosslinker-free PNIPAM particles were detectably larger than those around the crosslinked particles, possibly due to the larger free volume of the host polymer matrix. Interestingly, the SPR band positions were very similar, although the incorporated AuNPs were detectably different in size and distribution. This observation could possibly be due to the somewhat limited plasmonic coupling of the integrated AuNPs and their polydispersity. Specifically, the possibility of plasmonic couplings among the AuNPs in the crosslinker-free PNIPAM particles could be lower (a lower number of AuNPs spaced out in the polymer network). However, a higher number of AuNPs in the crosslinked polymer particles could easily undergo plasmonic couplings, resulting in the redshift of the SPR band. These unexpected SPR patterns are the components under investigation.

### 3.3. Physical Properties of Composite Particles Prepared from Crosslinker-Free and Crosslinked PNIPAM Particles

To ensure the successful integration of AuNPs, the SPR patterns of the composite particles were monitored as a function of temperature (Figure 3). Increasing the solution temperature above the LCST can readily deswell both composite particles to induce strong SPR couplings by the integrated AuNPs, resulting in the shift of the SPR band to a longer wavelength. In the case of the composite particles prepared with crosslinker-free PNIPAM particles, the notable change in the SPR band pattern was ~30 ℃ upon gradually increasing the solution temperature. The redshift of the plasmonic SPR bands (the dotted line in Figure 3a) caused by decreasing the diameter of the composite particles suggested the coupling of the integrated AuNPs. For the composite particles prepared with crosslinked PNIPAM particles, the SPR band changes (the dotted line in Figure 3b) were noticeable at slightly over 32 ℃, possibly implying that the presence of the crosslinker slightly raised the LCST of the polymer particles. As both composite particles exhibited reversible SPR patterns upon heating and cooling, it was clear that the *in situ*-formed AuNPs were successfully incorporated throughout the polymer particle network. The appearance of additional peaks at 970 nm for both composite particles should be associated with the O-H stretching vibration of water molecules at elevated temperatures, which has been explained by others [46,47]. This characteristic peak is a sensitive indicator for the local environment of water (i.e., the temperature and water binding state where the intensity increase and gradual blueshift indicate the increase of temperature). To quantitatively examine the total amount of integrated AuNPs, AA analysis was carried out after dissolving the AuNPs using a strong acid. Based on the calibration curve of Au atoms (Appendix A), the loading efficiency of the AuNPs was around 51% (0.51 mg/mL) for the crosslinker-free and 78% (0.78 mg/mL) for the crosslinked polymer particles. The importance of the crosslinker across the PNIPAM particle network to uptake a higher number of the guest AuNPs is evident, as these polymer particles do not possess special functional groups to interact with metal NPs (i.e., AuNPs). As these two polymer particles possessed very similar repeating networks, it was difficult to observe meaningfully different characteristics by additional analysis techniques (Fourier transform-infrared spectrometer, thermogravimetric analyzer, and powder x-ray diffractometer shown in Appendix A). 

Separately, the stability of these composite particles was monitored under various conditions (such as dispersion in alcohol, adding organic and inorganic bases, and adding salt). This test allowed for understanding the physical properties of the polymer matrix and/or colloidal characteristics of integrated AuNPs. Unlike bare AuNPs, resuspending both composite particles in an organic solvent (i.e., EtOH) remained the same without causing any aggregation and/or phase separation (Appendix A). Even the addition of salt (1 M KCl), strong inorganic base (1 M KOH), and organic base pyridine into the composite particles in water or EtOH did not show detectable changes. Adding these additives to unmodified colloidal metal NPs, including AuNPs, can easily induce severe agglomeration and/or aggregation. This simple test strongly suggested the high stability of the composite particles, where the host PNIPAM materials played an important role to effectively secure the colloidal stability of the integrated AuNPs for catalytic applications.

### 3.4. Catalytic Properties of Crosslinker-Free and Crosslinked Composite Particles in the Homocoupling Reaction of Phenylboronic Acid

These composite particles were then tested as catalysts in the homocoupling of phenylboronic acid as a function of time (Figure 4). The moderate heating of the reaction medium at 65 ℃ resulted in the rapid conversion of phenylboronic acid to biphenyl under aerobic conditions. This type of homocoupling reaction with high yields has been often employed to validate the catalytic activities of composite particles where the only variable was the degree of crosslinking across the polymer network [15,48,49,50,51,52,53]. When the equal volume of the prepared composite particles (i.e., 2.0 mL) was used in the reaction, the crosslinker-free composite particles showed relatively lower yields, presumably due to the detectably fewer number of integrated AuNPs discussed above. The amount of the integrated AuNPs was adjusted to that of the crosslinked composite particles, and it was evident that crosslinker-free composite particles resulted in a notably faster rate of reactions. This is because the loose PNIPAM network could offer a mass transfer environment for the reactants and products that can effectively interact with the physically integrated AuNPs. We also speculated that the crosslinker-free composite particles might not preserve a globular shape in EtOH in the absence of the BIS crosslinker, which could result in a somewhat disentangled polymeric layout (i.e., more liberated polymer chains with AuNPs). Alcohol solvents, including EtOH, favorably interact with the PNIPAM chain/network (e.g., hydrophobic isopropyl and ethyl groups, and amide and alcohol groups) to weaken the coil-globular aggregate structure (i.e., particle shape) [54,55,56]. As such, this structural change could allow for the easy access of organic species to the surface of integrated AuNPs during the catalytic reaction. This speculation was partially supported by the electron microscope images (Figure 5). The SEM image shows the general distribution of the crosslinker-free composite particles in EtOH, where many particles appeared to be flattened and lost their original shape. It was slightly difficult to find a globular, discrete sphere form of the composite particles during the TEM image analysis. Upon spreading out the crosslinker-free PNIPAM particles (e.g., possibly degenerated in EtOH), only the integrated AuNPs were visible, while the polymer network did not provide enough electronic contrast to obtain a particle shape. Unlike PNIPAM materials in water, it is reported that alcohol solvents are known to be good for interacting with the hydrophobic and hydrophilic segments of PNIPAM chains to eliminate the LCST feature (e.g., temperature-responsiveness) [54,56]. However, the crosslinked composite particles still maintained a distinct and well-defined particle shape in EtOH, as confirmed by the SEM and TEM images. Although the TEM images show a somewhat flattened form of the crosslinked composite particles, the traces of polymer particle shapes were visible as gray circles. Overall, the high catalytic activity of these composite particles in the C-C bond-forming reaction was possibly caused by the enhanced stability of the integrated AuNPs in nanoscale and the maintenance of a relatively large free volume across the polymer network.

Given these findings, it is evident that the crosslinker-free PNIPAM particles could serve as a highly porous host material to integrate guest AuNPs because the resulting composite particles maintained great stability to serve as a reactive catalyst. The loosely crosslinked network of the PNIPAM particles improved the mass transfer environment to induce a faster formation of AuNPs *in situ* and better catalytic reactivity. However, the crosslinker-free polymer particles possessed a slightly lower integration capacity for guest metal NPs and were somewhat structurally degenerated in EtOH. Additional homocoupling reactions of arylboronic acid derivatives and their recyclability using these composite particles are underway. To the best of our knowledge, this work clearly demonstrated the possibility of integrating guest metal NPs into a crosslinker-free polymer matrix to show their unexpectedly high stability and reactivity that can lead to the development of practical catalytic systems. 

## 4. Conclusions

The reliable formation of crosslinked and crosslinker-free PNIPAM particles was achieved by conventional radical polymerization in the presence and absence of the BIS crosslinker, respectively. When these polymer particles were used as host materials to integrate guest AuNPs via the light-induced reduction method, the crosslinker-free PNIPAM particles allowed for the faster formation of slightly larger and more polydisperse AuNPs. This phenomenon was possibly due to the higher diffusion rate of gold ions across the loose polymer network, enabling the faster nucleation and larger growth of AuNPs. Interestingly, the integration capacity of the guest AuNPs was slightly lower for the crosslinker-free PNIPAM particles while the overall stability was comparable to the crosslinked composite particles under various conditions. The subsequent use of the crosslinker-free composite particles, as a colloidal catalyst, displayed detectably lower catalytic activities in the homocoupling of phenylboronic acid in EtOH under the same polymer concentration. This result was possibly due to the relatively larger and more polydisperse as well as a fewer number of AuNPs across the polymer particles. Upon using an equal concentration of integrated AuNPs, the crosslinker-free composite particles showed much better catalytic activity than the crosslinked composite particles. This observation clearly suggests that the crosslinker-free PNIPAM particles with the loosely self-crosslinked network and more free volume readily enhanced the accessibility of organic species to the catalytic sites of the integrated AuNPs. As such, the polymer network density of the host materials mainly influenced the formation of the guest AuNPs and their overall catalytic functions. Precisely controlling the polymer network and examining the formation of metal nanoparticles along with the loading efficiency will lead to the development of highly reactive catalytic systems. 

## Figures and Tables

**Figure 1 nanomaterials-13-00416-f001:**
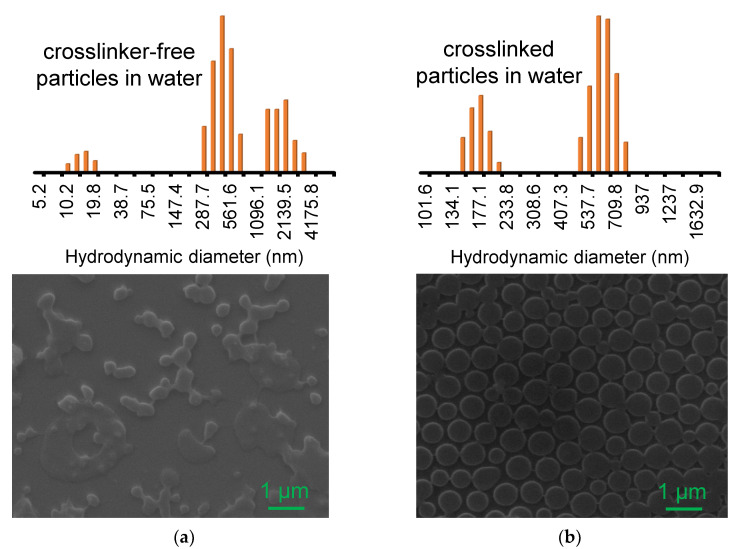
SEM image of (**a**) crosslinker-free and (**b**) crosslinked PNIPAM particles and their corresponding size distribution by DLS.

**Figure 2 nanomaterials-13-00416-f002:**
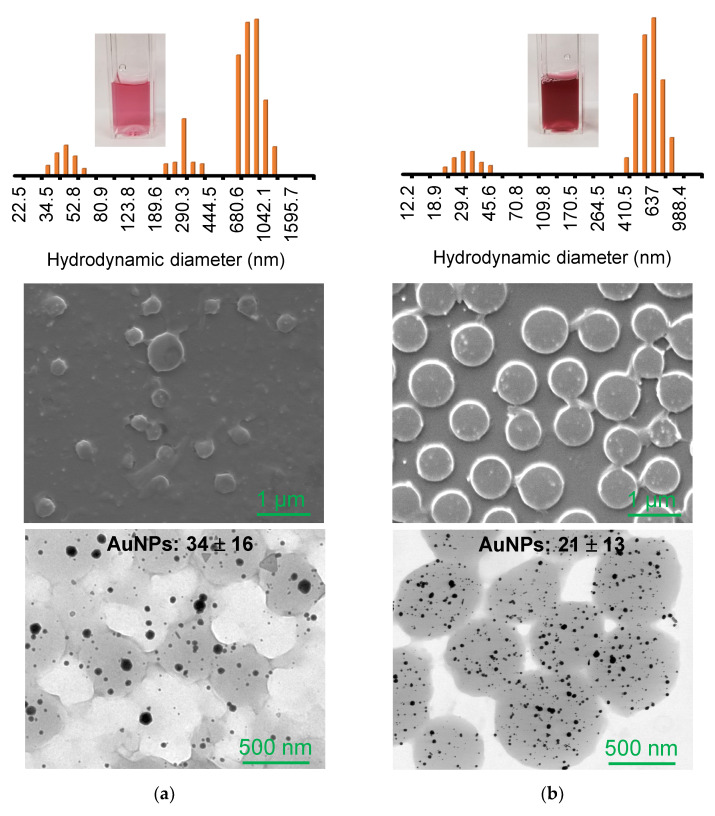
SEM and TEM images of composite particles prepared with (**a**) crosslinker-free and (**b**) crosslinked PNIPAM and their corresponding size distribution by DLS.

**Figure 3 nanomaterials-13-00416-f003:**
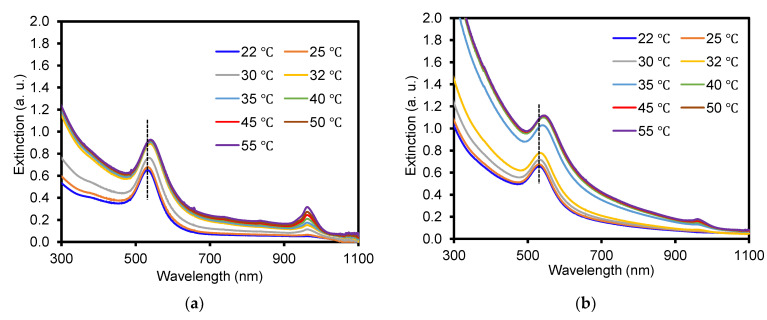
Absorption (i.e., SPR) patterns of composite particles prepared with (**a**) crosslinker-free and (**b**) crosslinked PNIPAM as a function of temperature (peak at ~970 nm is water).

**Figure 4 nanomaterials-13-00416-f004:**
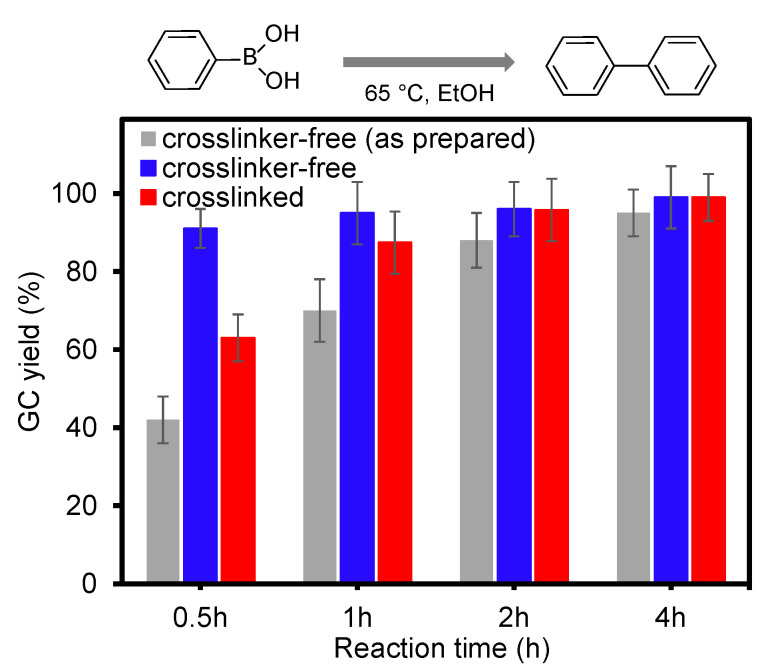
Homocoupling reaction yields (%) using both composite particles as a function of time [note: crosslinker-free (as prepared) indicates that the composite particles contain the fewer number of AuNPs across the crosslinker-free PNIPAM particles].

**Figure 5 nanomaterials-13-00416-f005:**
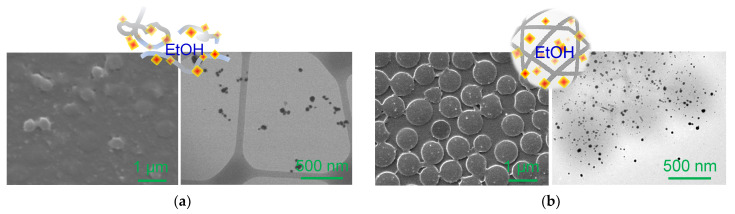
SEM and TEM images of composite particles prepared with (**a**) crosslinker-free and (**b**) crosslinked PNIPAM as well as their speculated structural features in EtOH.

## Data Availability

Not applicable.

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
