# Peer review of "Integration of Gold Nanoparticles into Crosslinker-Free Polymer Particles and Their Colloidal Catalytic Property"

_nanomaterials, 2023, doi:10.3390/nano13030416_

Round 1

Reviewer 1 Report

Jian Hou1, Bin Li et al. reported the “Integration of Gold Nanoparticles into Crosslinker-Free Polymer Particles and Their Colloidal Catalytic Property”. The work is a good contribution in to the field of catalysis. The paper is publishable in this journal but after major revision. This paper need a lot of improvement and missing characterization date which should be include in the revised manuscript.   

Comments and suggestion.

1.      What is the novelty of your work? Although a lot of literature is present on such type of work. Hybrid Cryogels is more preferred over conventional hydrogels for catalytic reduction purposes. Read the following literature and compared your work and improve the introduction part. (1). Silver and palladium nanoparticle embedded poly(n-isopropylacrylamide-co-2-acrylamido-2-methylpropane sulfonic acid) hybrid microgel catalyst with pH and temperature dependent catalytic activity. Korean J. Chem. Eng., 37(4), 614-622 (2020) (2) Hybrid cryogels composed of P(NIPAM-co-AMPS) and metal nanoparticles for rapid reduction of p-nitrophenol. Polymer 193 (2020) 122352. (3) Highly porous cryogels loaded with bimetallic nanoparticles as an efficient antimicrobial agent and catalyst for rapid reduction of water-soluble organic contaminants. Journal of Environmental Chemical Engineering 9 (2021) 106510.

2.      In materials part include the % purity and company name of the chemical. The sequence of the chemicals changes and start from monomer, cross-linker and etc.

3.      When using of reducing agent for the reduction no need of visible light irradiation. Why you use it. Give the solid reason. Very confusing. Read the following article for comparison. Synthesis of sensitive hybrid polymer microgels for catalytic reduction of organic pollutants. Journal of Environmental Chemical Engineering 4 (2016) 34923497.

4.      Mention the equilibrium swelling of cross-linked NIPAM microgel. In how much time it reached to its swelling equilibrium because it’s very important for the catalysis application.

5.      With increase in temperature there must be some aggregation in nanoparticles and NIPAM will also shrink and chance that show red shift in UV spectra but I have not observed it in Figure 3a and 3b. Give the solid reason.

6.      The interaction mechanism should include in the revised manuscript. That how Gold nanoparticles interact with NIPAM network.

7.      FTIR, XRD, TGA and EDS should include in the revised manuscript because with the present data the paper is not publishable in nanomaterials journal.

8.      SEM and TEM images is not enough. It’s a comparative study and need more clear SEM and TEM images. SEM and TEM should be explain separately. Don’t combined SEM and TEM images.

9.      Improve the conclusion part.

Reviewer 2 Report

Kim and coworkers prepared the incorporation of gold nanoparticles (AuNPs) into crosslinker-free PNIPAM, and this kind of AuNPs had larger and more polydisperse and slightly inferior integration efficiency. Notably, the the crosslinker-free AuNPs as a colloidal catalyst display a higher reactivity toward the homocoupling reaction. Therefore, the present work give some hits for the development of highly practical catalytic systems. However, as is metioned by the authors, various  AuNPs  materials have been prepared and studied by this group, while appication about the coupling reactions is still limited in the self-coupling of  phenylboronic acid. In my opinion, the author should display the advantages of the present materials over the conventional homogeneous catalysis.

Round 2

Reviewer 1 Report

The author addressed all the issues clearly and I am recommended this manuscript for publication.

Reviewer 2 Report

Accept in present form